# Femtosecond concerted rotation of molecules on a 2D material interface

Kiana Baumgärtner[1], Misa Nozaki[2], Marvin Reuner[3], Nils Wind[4,5], Masato Haniuda[2], Christian Metzger[1], Michael Heber [5], Dmytro Kutnyakhov [5], Federico Pressacco [5], Lukas Wenthaus [5], Keisuke Hara [2], Kalyani Chordiya[3,6], Chul-Hee Min[7], Martin Beye [5], Friedrich Reinert [1], Friedrich Roth [8,9], Sanjoy Kr Mahatha[5,10], Anders Madsen [11], Tim Wehling[3,6], Kaori Niki [2], Daria Popova-Gorelova [3,6,12], Kai Rossnagel [7,13,14] & Markus Scholz [5] ✉

Interfaces between molecules and 2D materials exhibit energy-driven functionalities, wherein charge transfer directs molecular motion. Unlike equilibrium systems, where molecular assemblies settle into static configurations, continuous energy input can drive transient, collective molecular rearrangements. Here, we reveal ultrafast spectroscopic fingerprints of a collective rotational response of molecules on a 2D material following photoexcitation. Our results suggest that photoinduced charge transfer reshapes the interfacial energy potential, giving rise to macroscopic, unidirectional molecular rotation and the formation of a homochiral molecular arrangement. Using a multiplexed ultrafast photoemission spectroscopy approach, we simultaneously track electronic states, atomic positions, and orbital wavefunctions with femtosecond and sub-Ångström resolution. Multimodal valence and core electron emission analysis disentangles the intertwined electronic-structural dynamics of the molecule and the 2D material, revealing the dynamic modulation of charge distribution and intermolecular forces that drive collective molecular motion. Our findings open a pathway for designing energy-driven molecular systems with tunable interfacial dynamics, with potential applications in chiral engineering and active matter systems.

Understanding and controlling molecular motion at interfaces is fundamental to designing functional materials with tailored properties, particularly in molecular electronics, heterogeneous catalysis, and chiral engineering[1–6]. In addition, ratchet mechanisms convert random thermal fluctuations into directed motion under non-equilibrium conditions, as exemplified by light-driven rotors and field-controlled molecular gears but also in larger networks that enable autonomous processes for applications ranging from information storage to nanoscale actuators[7–9].

The structural organization of molecular films is dictated by a competition between molecule-substrate interactions, such as covalent bonding and charge transfer, and intermolecular forces, including Coulomb, van der Waals, and dipolar interactions[10,11]. These forces collectively shape the potential energy surface, which, under equilibrium conditions, determines the molecular configurations that molecules adopt in long-range ordered films.

External stimuli, such as optical excitation or electric fields, can transiently modify the intermolecular and molecule-substrate interactions, as well as the potential energy surface, enabling rearrangement processes that would otherwise be inaccessible under equilibrium conditions[4,12]. As a result, molecular assemblies can undergo phase transitions, reversible self-assembly, or directed

---

**Fig. 1 | Time-resolved orbital tomography and band structure evolution at the CuPc/TiSe₂ interface. a** Conceptual sketch of the experiment including a real dataset of the valence-band and core-level emissions of CuPc/TiSe₂. These four-dimensional data sets capture ultrafast dynamics of the adsorbed molecules on a sub-picosecond timescale, resolving changes in energy-momentum space. **b** Momentum-integrated valence-band region. At four delay times $t_{-1}$, $t_0$, $t_1$ and $t_2$ the momentum maps at selected energies are extracted: **c** CuPc LUMO, (**d**) Ti $3d$ conduction band of TiSe₂, (**e**) CuPc HOMO. **f** Corresponding energy-momentum maps of the valence-electron region dynamics. For (**c**–**e**), we integrated in energy over ±220 meV around the denoted center energy. For (**c**), (**e**), we applied a Gaussian filter with a kernel size of 5 × 5. For (**d**), (**f**), the kernel size is 2 × 2. The red and white colored arrows in (**c**) indicate the projected light incidence (315°) for all measurements. Source data are provided as a Source Data file[50].

motion, unlocking dynamic and adaptive behaviors that mimic nature's efficiency in transient systems[9,12,13]. These dynamic behaviors go beyond static equilibrium configurations, enabling systems to perform tasks that would be impossible in a stable state.

Here, we demonstrate how charge-driven potential energy surface modification at a hybrid molecule-2D material interface induces ultrafast, synchronized, unidirectional molecular rotation. Using a multiplexed time-resolved photoemission spectroscopy approach[14–19], we track atomic positions[20,21], charge dynamics, and orbital evolution with femtosecond resolution[22–24], revealing the atomistic mechanisms that drive collective molecular motion at the nanoscale. Our results further suggest that the interplay between electronic and structural dynamics leads to the spontaneous formation of homochiral molecular domains, an essential step toward realizing controlled chiral molecular architectures. By integrating experimental observations with theoretical modeling, we establish a framework for engineering chiral-selective molecular motion at surfaces.

The interface we study is formed by copper(II)phthalocyanine (CuPc) molecules in a self-assembled, highly ordered molecular film adsorbed on the layered transition-metal dichalcogenide TiSe₂ (see "Sample preparation" and Supplementary Note 1.1). To directly correlate the photoinduced charge and energy flow across the interface with the molecular structural response, we have implemented a multiplexed electronic movie approach, integrating four modalities of time-resolved photoelectron spectroscopy: time-resolved orbital tomography (trOT), time- and angle-resolved photoemission spectroscopy (trARPES), time-resolved X-ray photoelectron spectroscopy (trXPS), and time-resolved X-ray photoelectron diffraction (trXPD). These techniques use ultrashort-pulsed extreme ultraviolet and soft x-ray radiation for combined momentum-dependent valence-electron

and atomic site-specific core-electron emissions, as illustrated in Fig. 1a.

## Results

### Charge transfer dynamics

Upon optical pumping of the hybrid interface (at time $t_0$), the most prominent effect observed in the momentum-integrated trARPES data in Fig. 1b is the redistribution of spectral weight from the Se $4p$ valence band to the Ti $3d$ conduction band[14,15]. Subsequently, hot electrons relax by intraband scattering to lower energies into the Ti $3d$ conduction band minimum, while hot holes scatter to higher energies into the Se $4p$ valence band maximum.

From the trARPES intensity momentum distribution, we can identify the lowest unoccupied molecular orbitals (LUMO and LUMO', Fig. 1c). These orbitals are transiently populated at $t_0$, but play only a minor role in modulating the interfacial potential landscape, as only about 1% of the molecules are excited to this state. In the iso-energy cuts at about 180 meV above and 300 meV below the Fermi level, we identify the Ti $3d$ conduction band (Fig. 1d) and the CuPc highest occupied molecular orbital (HOMO, Fig. 1e), respectively. Our calculations (see Supplementary Note 2.2) show that the momentum distribution of the HOMO for multiple domains is represented by a ring-like intensity modulation around the Γ point with a radius of about $(1.67 \pm 0.01)$ Å⁻¹.

Experimentally, we find that the TiSe₂ conduction-band population reaches maximum intensity later for CuPc/TiSe₂ ($t_{max} = 0.369 \pm 0.041$ ps) than for pristine TiSe₂ ($t_{max} = 0.181 \pm 0.054$ ps). The corresponding decay constants are $\tau = 0.441 \pm 0.021$ ps and $\tau = 0.386 \pm 0.054$ ps, respectively; the decay constant is slightly longer for the CuPc-covered surface, but this

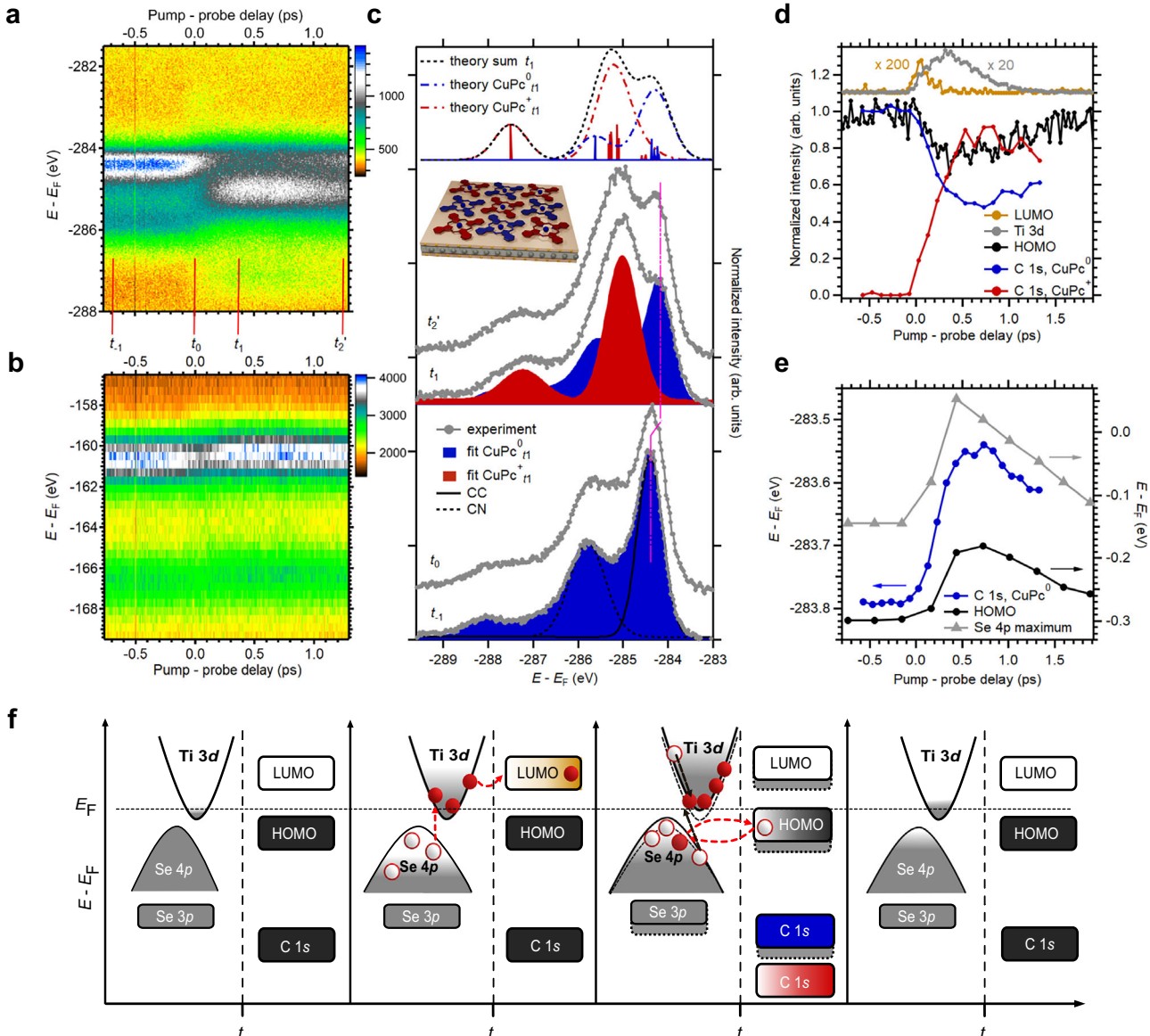

**Fig. 2 | Interfacial electronic dynamics. a** Transient trXPS spectrum of the C 1s core level and (**b**) Se 3p core level. **c** C 1s trXPS spectra at $t_{-1}$, $t_0$, $t_1$ and $t_2'$ (from bottom to top, marked in (**a**)) including fits of the signatures of neutral (blue) and charged (red) molecules and the corresponding simulations (dashed lines). The inset depicts a probable arrangement of both neutral and charged molecules. **d** Transient normalized photoemission intensities of the CuPc LUMO (light brown), the HOMO (black), the C 1s core level of neutral (blue) and charged (red) CuPc molecules, and the Ti 3d states of the substrate (gray). **e** Transient energies of the C 1s level of neutral molecules (blue), the HOMO (black), and the Se 4p valence-band maximum of the substrate (gray). **f** Sketch of the charge-transfer dynamics between CuPc and TiSe$_2$. Source data are provided as a Source Data file[50].

difference is within the combined uncertainty (Supplementary Note 1.7). The shift of $t_{max}$ to longer delays indicates that the excited-state population evolves through an intermediate stage before relaxation. We attribute this behavior to an extended cascade of secondary hot carriers generated by inverse Auger (impact-ionization) processes[14,15], which are prolonged further by charge exchange with the CuPc layer. This observation highlights the critical role of the molecular environment in shaping the dynamics of 2D materials, effectively creating a hybrid system with modified electronic properties.

To determine the role of hot hole transfer in modulating the interfacial potential, we quantify the ratio of $CuPc_{t_1}^+$ to all molecules at $t_{-1}$ ($CuPc_{t_{-1}}$) using trXPS (Fig. 2a)[21,25]. First, we identify the spectral signatures of $CuPc_{t_1}^+$ (red) and $CuPc_{t_1}^0$ (blue) molecules in the carbon 1s (C 1s) core-level signal by comparison with calculations (see "Methods"

and Supplementary Note 2.3.3) (Fig. 2c). We find that the $CuPc_{t_1}^+$ spectrum shifts to higher binding energies and exhibits a modified spectral signature compared to the spectrum before excitation ($CuPc_{t_{-1}}$), due to reduced core-hole screening during hot-hole transfer from the Se 4p band to the HOMO (see "Methods" and Supplementary Note 2.3.3). In contrast, the $CuPc_{t_1}^0$ signature shifts rigidly to lower binding energies due to surface potential modulation (see "Methods" and Supplementary Note 2.1). From the ratio of the $CuPc_{t_1}^+$ / $CuPc_{t_{-1}}$ peak intensities, we estimate that approximately 45% of the molecules are charged. We note that the C 1s trXPS peak shifts and intensities correlate well with the dynamics of the characteristic trARPES signatures (Fig. 2d, e).

Due to the proximity of the CuPc HOMO to the Fermi level $E_F$, hot holes are injected into the molecule within ~375 fs after absorption of the pump pulse ($t_I$). This charge transfer process alters the

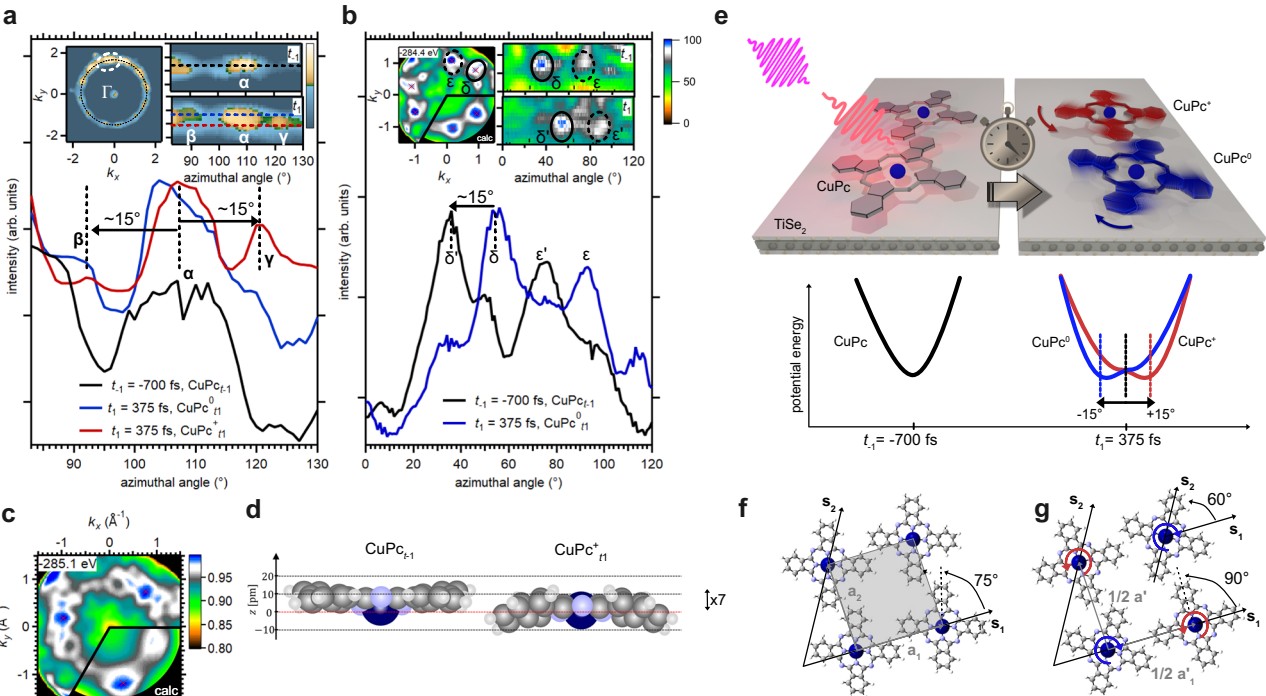

**Fig. 3 | Molecular structural dynamics. a** Structural dynamics extracted from the CuPc HOMO momentum distribution. Azimuthal line cuts along the dashed lines marked in the insets show the appearance of additional features (β, γ) at $t_1$, separated by −−15°/ + 15° from the HOMO features (α) at $t_{-1}$. The integration range of the azimuthal line cuts is 0.1 Å$^{-1}$. **b** Analysis of the C 1$s$ XPD pattern of the neutral CuPc$^0_{t_1}$ molecules. The signatures are indicated by red crosses. At $t_1$, the signatures associated with neutral CuPc exhibit an azimuthal shift corresponding to a clockwise rotation of approximately −15°, as deduced from the displacement of the δ → δ′ and ε → ε′ features. The azimuthal line cuts were extracted at $|k| = (0.7 ± 0.15)$ Å$^{-1}$. For $t_{-1}$, the energy window spans the (282.6 eV − 284.1 eV) binding energy. At $t_1$, the XPD was extracted between (284.1−284.8 eV) and (282.7−283.6 eV) binding energy for charged and neutral molecules, respectively. **c** Comparison of the C 1$s$ XPD pattern of CuPc$^+_{t_1}$ with theory. The three red crosses mark the characteristic signatures,

whose azimuthal displacement is consistent with a rotation of approximately +15. **d** Structural models extracted from the XPD pattern for CuPc and CuPc$^+_{t_1}$. Atomic diameters are scaled by a factor of 1/7. **e** Schematic representation of the experimental results of (**a**), (**b**) and (**c**): cogwheel rotational motion of the adsorbed molecules on a sub-picosecond time scale in energy-momentum space and real space. The rotations are triggered by changes in the potential energy surface following charge carrier transfer between TiSe$_2$ and CuPc and are in opposite directions for charged (red) and neutral (blue) molecules. Panel (**f**) and (**g**) displays schematic representations of the molecular arrangements at $t_{-1}$ and $t_1$, respectively. Panel (**f**) is based on data extracted from LEED measurements, whereas panel (**g**) derives from pair-potential calculations. Source data are provided as a Source Data file[50].

electrostatic potential at the interface. We model the excitation-induced binding energy shifts using a dielectric approach that assumes charge transfer from TiSe$_2$ to a fraction of the CuPc molecules, resulting in an electrostatic potential that qualitatively reproduces the experimental level shifts (see "Methods" and Supplementary Note 2.1). Consequently, all energy levels shift toward lower binding energies, reaching a maximum shift of -200 meV at $t_1$ (Fig. 1f, Fig. 2a, b). The charge carrier dynamics, as deduced from the momentum maps (Fig. 1b–f), core-level shifts (Fig. 2a, b), and intensity transients (Fig. 2d, e), are schematically summarized in Fig. 2f.

### Molecular rotation and deformation

As shown in Fig. 3a, the HOMO electron density distribution forms a ring-like structure with 12 distinct peaks (α) in momentum space. At $t_1$, we observe additional features (β and γ) separated from the initial peaks by about −15° ± 3° and +15° ± 3°, respectively, indicating rotational motion of the molecules. While the trOT data shows the occurrence and magnitude of the molecular rotation, it does not provide directional information or distinguish between charged and neutral molecules.

Similar rotations are inferred from the C 1$s$ trXPD patterns (Fig. 3b). However, the distinct spectral fingerprints of neutral and charged molecules in the C 1$s$ trXPS spectra, together with a trXPD R-factor analysis, consideration of steric constraints, and pair-potential calculations, allow us to determine the direction of rotation and disentangle the atomic rearrangement for each species.

The C 1$s$ trXPD patterns of neutral and charged molecules display 6 and 3 distinct peaks, respectively, due to intramolecular scattering. Figure 3b shows the trXPD pattern and the extracted neutral molecule contributions. At $t_{-1}$, the neutral molecule signal is integrated between −282.6 eV and −284.1 eV. At $t_1$, the neutral and charged molecules are isolated by integrating the C 1$s$ signal between −284.1 eV and −284.8 eV (neutral) and between −282.7 eV and −283.6 eV (charged), respectively.

Azimuthal line scans extracted at $(0.7 ± 0.15)$ Å$^{-1}$ from the trXPD data at $t_{-1}$ and $t_1$ (Fig. 3b) show that the δ and ε features of the neutral molecules rotate clockwise by −15° ± 3°, which is consistent with the R-factor analysis (Supplementary Note 1.6). In addition, the δ′ and ε′ appear shifted toward the Γ point for CuPc$^0_{t_1}$, indicating a decreased adsorption height. Multiple scattering calculations for different model geometries reveal out-of-plane deformation with benzene wings approaching the substrate (Fig. 3d).

For the charged molecules, R-factor analysis does not yield an unambiguous rotational angle within the error margin (Supplementary Note 1.6). However, steric constraints must also be considered. At a coverage of one monolayer, the molecules are densely packed. In such an arrangement, a rotation of the neutral molecules necessarily implies a correlated motion of neighboring molecules in opposite directions to avoid collisions. The measured XPD pattern and the pattern calculated for CuPc$^+_{t_1}$ molecules rotated CCW by +15° ± 3° are shown in Fig. 3c.

To further understand the observed rotations, we performed pair-potential calculations to evaluate how charge transfer modified the

intermolecular interaction landscape. We calculated the pair potential between the molecules using the partial charges on each atom in the molecules obtained from the RASCI calculation[26] of the ground state of $CuPc_{t_{-1}}$ and the ionized state of $CuPc_{t_1}^+$ (see Supplementary Note 2.4). The potentials are shown in the lower part of Fig. 3e. This method, previously applied to similar systems[27–29], shows that, depending on the initial in-plane orientation of the molecules relative to their unit cell (angles marked in Fig. 3f, g), the $CuPc_{t_1}^+$ and $CuPc_{t_1}^0$ molecules adapt to the new potential energy surface by rotating by $+15°$ and $−15°$, respectively, in agreement with the experimental results. The result is illustrated in Fig. 3e.

This ultrafast rotation is captured in both tomographic trOT snapshots of the HOMO (Fig. 3a) and trXPD patterns of the C 1s core level (Fig. 3b, c) within ~375 fs after absorption of the pump pulse. However, not all molecules rotate at $t_1$; signatures of non-rotated molecules are detected in both the HOMO snapshots and the trXPD pattern. About 20% of the neutral molecules remain unrotated, and a similar fraction is estimated for charged species, with a $\pm 10\%$ margin of error due to pattern overlap.

### Homochiral domain formation

Our combined trOT and trXPD data provide consistent and complementary electronic (orbital) and structural (atomic-site) perspectives on the uniform, unidirectional rotation of the majority of charged and neutral molecules within the probed area. However, static low-energy electron diffraction (LEED) revealed mirror-symmetric domains in the as-prepared CuPc/TiSe₂ monolayer (see Supplementary Note 1.1), and the pair-potential calculations predict that molecules in mirror domains should rotate in opposite directions. Yet, within the sensitivity of the trXPD experiment, we observe no signatures consistent with counter-rotating domains under continuous photoexcitation.

The absence of opposite-rotation features implies that a homochiral molecular configuration emerges under photoexcitation. This behavior is consistent with that of other energy-driven molecular systems on surfaces[30–34]. The homochiral arrangement vanishes once the excitation is removed, confirming that the effect is reversible and not due to permanent structural modification.

We attribute the transient homochirality to a photoinduced rebalancing of the molecule-substrate and intermolecular forces. When the energy barrier between mirror-symmetric domains is small, the system can overcome it under external excitation, favoring the formation of a single chiral phase[30,31]. The removal of domain walls reduces the overall energy dissipation, analogous to non-equilibrium Ostwald ripening[35]. This transient stabilization is further supported by the out-of-plane deformation of $CuPc_{t_1}^+$, which breaks the fourfold symmetry of neutral molecules into a twofold one[36–38]. The nearly square molecular unit cell (Fig. 3f) suggests a small potential barrier between the domain and the mirror domain, consistent with the transient suppression of the mirror domains during pumping. We therefore propose that the interfacial potential modulation drives the symmetry breaking and domain coarsening via non-equilibrium processes. The molecular arrangements deduced from LEED experiments and pair-potential calculations are shown in Fig. 3f–g.

## Discussion

In summary, we have observed ultrafast concerted rotation of achiral molecules on a 2D material driven by charge carrier transfer between substrate and molecules. Our results demonstrate how non-equilibrium conditions can significantly alter the molecular assembly process by shifting the balance between intermolecular forces and molecule-substrate interactions. Under non-equilibrium conditions, continuous energy input reshapes the energy landscape, allowing molecular systems to overcome local barriers and form dynamic configurations that are unattainable in equilibrium. This energy-driven process initiates femtosecond-scale unidirectional rotation, which stabilizes a more ordered, homochiral state by minimizing domain wall formation. Prolonged relaxation times in TiSe₂ due to molecular adsorption suggest that the molecular layer modifies charge carrier dynamics, offering potential for controlling energy transfer and charge transport in molecular electronics, chiral systems, and spintronics.

Our multimodal approach integrating trOT, trARPES, trXPS, and trXPD provides comprehensive insight into both global interfacial electronic dynamics and local atomic site-specific charge-state dynamics. Direct, microscopic, and multi-perspective insight into the interplay between structural and electronic dynamics is crucial for better understanding and controlling molecular motion on surfaces. Our findings open a path for the rational design of new functionalities in hybrid systems, particularly in non-equilibrium active matter. This could lead to innovations in molecular machines, chiral engineering, and smart material systems, where dynamic control of molecular behavior is essential.

## Methods

### Photoemission experiments

The time-resolved CuPc/TiSe₂ photoemission measurements were performed at both a HHG and a FEL source, with a pump excitation energy of 1.6 eV and a probe pulse photon energy of 36.3 eV (both p-polarized). Both experiments lead to the same observed dynamics, but due to longer acquisition times and thus better statistics, we have chosen to present the valence-band data obtained at the HHG source in the main manuscript[39]. The C 1s core-level measurements were performed at the PG2 beamline at FLASH in Hamburg, Germany, with a pump photon energy of 1.6 eV and a probe photon energy of 370 eV. The pump and probe pulses are incident on the sample at a polar angle of 68° and an azimuthal angle of 0° with respect to the Γ-M direction of the substrate. The spot sizes of the pump and probe beams on the sample are approximately $(260 \times 150)$ μm² and $(200 \times 100)$ μm², respectively, and both beams are aligned to spatially overlap. The average FEL pulse energy of 30 μJ is attenuated by several thin-film filter foils and nitrogen gas. The pump laser delivers a maximum fluence of 1 mJ/cm². The energy resolution of the experiment is 80 meV, evaluated by fitting the Fermi edge of an Ag(110) crystal at room temperature. The temporal resolution is $(95 \pm 5)$ fs for the valence band data taken at the HHG source and $(180 \pm 10)$ fs for the core-level data taken at the FEL source, evaluated from the cross-correlation of the pulses. The photoelectrons are detected by a time-of-flight momentum microscope[24]. The data sets, sampling over ~2 ps, were acquired in ~15 h. The energy of the obtained spectra is calibrated by comparing the central CuPc HOMO energy with a value previously measured at a He-I source using a hemispherical analyzer (Scienta Omicron R3000). The momentum maps are similarly calibrated at the Γ points of a clean TiSe₂ sample. All shown momentum maps are integrated over $\pm 220$ meV around the indicated center energy.

### Sample preparation

The CuPc monolayer on TiSe₂ was prepared in situ at a base pressure of $10^{-10}$ mbar. The TiSe₂ substrate of ~$(5 \times 5)$ mm² size was cleaved in situ and characterized by low-energy electron diffraction (LEED) and photoemission measurements prior to deposition of the molecules. CuPc molecules (gradient sublimated) were deposited by organic molecular beam epitaxy from a home-built Knudsen cell evaporator at a Knudsen cell temperature of 400 K and a deposition rate of 1 monolayer per 40 min at an evaporator-to-sample distance of 15 cm. These growth parameters were previously determined by means of LEED and photoemission measurements on CuPc molecules deposited on an Ag(110) crystal with a well-characterized growth. The LEED image characteristic of a monolayer of CuPc molecules on TiSe₂ is shown in the Supplementary Note 1.1 and a molecular superstructure of (2,4.5/4,0.2) is determined. This epitaxial relationship corresponds to a

point-on-line superstructure: the CuPc overlayer is commensurate with the TiSe$_2$ substrate along one crystallographic direction, while the perpendicular direction exhibits a quasi-incommensurate registry. From this superstructure, a real-space arrangement of densely packed flat-lying CuPc molecules with an intermolecular spacing of 13.8 Å can be determined. During the beamtime, several samples were prepared and their quality was verified by LEED.

## Time-resolved orbital tomography analysis

Centered at an energy of −310 meV below the Fermi level, we identify the CuPc HOMO from its intensity distribution in momentum space. The orbital is characterized by an intensity modulated ring centered around the Γ point with a radius of $(1.67 \pm 0.01)$ Å$^{-1}$. The energy integration range in Fig. 1c–e is ±220 meV around the denoted center energy. The calculation of a CuPc molecule adsorbed on an 8x8x1 TiSe$_2$ supercell is performed based on DFT calculations using the Vienna Ab Initio Software Package (VASP). Ground-state and excited-states calculations of a neutral and positively charged isolated CuPc molecule were performed with the MOLCAS package[40] using the RASCI method[26]. Further details of the calculation are given in the Supplementary Note 2.3.

At $t_0$ and between 550 and 990 meV above the Fermi level, we identify the CuPc LUMO, which is depopulated with a decay time of $\tau_{LUMO} = (92 \pm 50)$ fs. Its momentum distribution (Fig. 1c) shows an intensity close to the Γ point originating from Ti 3$d$ states as well as a ring-shaped intensity centered around the Γ point with a radius of $(1.60 \pm 0.05)$ Å$^{-1}$. This is supported by comparison with the calculated momentum distributions of the lowest unoccupied molecular orbitals (LUMO and LUMO') of isolated CuPc, which are superimposed on the experimental data as 1/2 LUMO + $\sqrt{3}/2$ LUMO'. Two excitation mechanisms can play a role in the transient LUMO population: a direct excitation from the HOMO to the LUMO, corresponding to an optical gap of ~1.1 eV, which is slightly smaller than the 1.4 eV reported for thin films on a metal substrate[41]. Due to the off-resonant excitation and the momentum distribution of the (1/2 LUMO + $\sqrt{3}/2$ LUMO'), we consider the most likely excitation channel to be via hot electron scattering from the Ti 3$d$ band into the LUMO. More details on the LUMO calculations are given in the Supplementary Note 2.3.4.

## Time-resolved XPS analysis

The C 1$s$ core-level spectrum of the neutral CuPc molecule on TiSe$_2$ consists of two main peaks and their shake-up satellites. We have performed calculations of the C 1$s$ XPS spectra of neutral CuPc and positively charged CuPc$^+$ using the extended Koopmans theorem[42]. The calculation reproduces the spectrum without the shake-up satellites, but overestimates the separation between the peaks. From the calculation of neutral CuPc, we determine that the lower binding energy peak originates from the carbon atoms in the benzene rings, while the higher binding energy peak originates from the carbon atoms in the pyrrole rings, in agreement with the literature[43]. The calculated C 1$s$ spectrum of positively charged CuPc$^+$ also has two peaks, but the structure of the spectrum changes. The separation between the benzene peak and the pyrrole peak increases by 1 eV. In addition, the benzene peak becomes broader and less intense. The calculated hole density of CuPc$^+$ is mainly located on carbon atoms in the pyrrole rings. These atoms experience a greater decrease in Coulomb screening than atoms in the benzene rings. This results in the larger shift of the pyrrole peak compared to the shift of the benzene peak toward higher binding energies. A small fraction of the positive charge is also located on the carbon atoms in the benzene rings, but is unevenly distributed over these atoms. As a result, the carbon atoms in the benzene rings experience a slightly different decrease in Coulomb screening, so that the corresponding peak becomes broader.

The C 1$s$ spectra after time zero ($t_0$) contain the signatures of both neutral and positively charged molecules. To determine the ratio of the contributions of the neutral CuPc and the positively charged CuPc$^+$ to the total spectrum, we fitted the total spectrum by adding the two calculated spectra in different ratios. For better comparison, we fitted the global positions of the spectra and reduced the overestimated peak separation by the same value for both spectra. We find that the total spectrum is best reproduced when about 45% of the positively charged molecules contribute to the spectrum.

## Time-resolved XPD calculations and analysis

The XPD data in Fig. 3b–c show the C 1$s$ signal of CuPc/TiSe$_2$ after subtraction of the inelastic background and correction for inhomogeneity of the 2D instrumental response function. To extract the molecular geometries at $t_{-1}$ and at $t_1$, the experimental momentum maps are compared with a multiple scattering calculation based on Fermi's Golden Rule using the Lippman-Schwinger equation for the final state[44] of 42 different sample geometries. Both intramolecular scattering within CuPc and scattering between CuPc and the substrate are considered. The atomic orbitals of the C atoms representing the initial states are calculated by Gaussian[45]. In the partially overlapping trXPS spectral signature of CuPc before time zero, CuPc$^+_{t_1}$ and CuPc$^0_{t_1}$, we can clearly isolate the intensities originating from the carbon atoms located in the benzene rings. Consequently, we extracted the experimental XPD pattern and calculated XPD momentum maps for these atoms. Due to the limited statistics in trXPD, we carefully three-fold symmetrized the experimental data. The quantitative agreement between the experimental and calculated XPD patterns is evaluated by means of a R-factor analysis according to the procedure shown in ref. 19. The 42 evaluated R-factors are shown in the Supplementary Note 1.6, and the models corresponding to the smallest R-factor are considered as the closest fits to the experiment (shown in Fig. 3b, c).

The simulated models consider a CuPc molecule adsorbed on a cluster of 58 atoms of 1$T$-TiSe$_2$ in its normal phase. The different geometric models of CuPc$^+_{t_1}$ and CuPc on TiSe$_2$ include different molecular out-of-plane distortions, adsorption heights, adsorption sites, and molecular in-plane rotations. For each model, three molecular orientations along the substrate high-symmetry directions according to the point-on-line growth are considered and their intensities are added to the resulting XPD momentum map. Further details of the calculations are given in the Supplementary Note 1.6.

## Dielectric model of the photoexcited system

The excitation-induced binding energy shifts of the CuPc- and TiSe$_2$-derived states toward $E_F$ are modeled by a dielectric model. We assume that a hole is transferred from TiSe$_2$ to a certain fraction of CuPc molecules. This fraction will be positively charged, while the rest of the molecules will remain neutral. The substrate carries the opposite negative charge. To describe the dielectric screening in this setup, the CuPc molecules are represented as a dielectric layer ($\varepsilon_1 = 4$[46]) with thickness h = 3.95 Å. The TiSe$_2$ substrate is modeled as a half-space with dielectric constant $\varepsilon_2$, while we assume vacuum above the CuPc layer ($\varepsilon_3 = 1$). For the electron-doped substrate, we consider $\varepsilon_2 = 60$[47] and, most simply, a perfect metal $\varepsilon_2 = \infty$. The CuPc molecules are assumed to form a square lattice with lattice constant a = 14.5 Å, and the positively charged molecules are modeled as a Gaussian surface charge density of 5 Å width, resulting in a total charge of +1e. The resulting electrostatic potential is calculated following refs. 48,49. and qualitatively captures the excitation-induced relative level shifts observed in the experiment. Further details of the calculation are given in the Supplementary Note 2.1.

## Theoretical description of the molecular in-plane orientations

Pair-potential calculations[27] were performed in order to determine the in-plane orientations of the molecules at $t_{-1}$ and $t_1$. A 2D layer of CuPc molecules is modeled based on the molecular superstructure

determined by LEED. The potential in the molecular layer is calculated taking into account intramolecular van der Waals and Coulomb interactions. By gradually varying the relative in-plane orientation between a central molecule and its surrounding neighbors, the molecular arrangement corresponding to the minimum pair potential is found. Further details of the calculations are given in the Supplementary Note 2.4.

## Data availability

The source data are provided with this paper. Data is available in the Zenodo repository under accession https://doi.org/10.5281/zenodo.17975297 Source data are provided with this paper.

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

## Acknowledgements

We thank DESY (Hamburg, Germany), a member of the Helmholtz Association HGF, for the provision of experimental facilities. Parts of this research were carried out at FLASH using beamline PG2. Beamtime was allocated for proposal 20190734. The momentum microscope at beamline PG2 is partially funded by the ErUM-Pro programme (grant numbers 05K22FK2 and 05K25FK2) of the German Federal Ministry of Research, Technology, and Space (BMFTR). K.B., C.M. and F.Re. acknowledge financial support of the Deutsche Forschungsgemeinschaft (DFG) through the Würzburg-Dresden Cluster of Excellence ctd.qmat (EXC 2147, project id 390858490). M.R. and D.P.-G. acknowledge the funding from the Volkswagen Foundation, project ID 96237. K.C. acknowledges funding from the Cluster of Excellence 'CUI: Advanced Imaging of Matter' of the Deutsche Forschungsgemeinschaft (DFG) - EXC 2056 - project ID 390715994. K.N. acknowledges the student dispatch program by the Japan Student Services Organization (JASSO). M.N., M.H. and K.N. thank the Supercomputer Center, the Institute for Solid State Physics and the University of Tokyo for the use of their facilities and acknowledge financial support from the Grants-in-Aid for Scientific Research (C) (No. 20K05643) from Japan Society for the Promotion of Science and the Initiative for Realizing Diversity in the Research Environment of Chiba University. N.W., M.H., D.K., S.M. and K.R. acknowledge financial support from the DFG (SFB 925, project ID 170620586). F.Ro acknowledges financial support by BMFTR (Grant No. 05K22OF2 within the ErUM-Pro programme). The authors would like to thank Holger Meyer and Sven Gieschen for instrumentation support, and Bernhard Mahlmeister for his contribution to Fig. 3e.

## Author contributions

M.S. designed the experiment. M.S., K.B., C.M. performed the data analysis. M.N., M.H., M.R., K.C., K.H., D.P.-G., T.W., K.N. performed the theoretical study. M.S., K.B., N.W., M.H., D.K., F.P., L.W., F.Ro, C.H.M. and S.M. conducted the experiment. M.S., K.B., and K.R. wrote the manuscript. M.S., K.B., M.N., M.R., N.W., M.H., D.K., F.P., L.W., K.H., K.C., C.H.M., M.B., F.Re, F.Ro, A.M., M.B., S.M., T.W., K.N., D.P.-G. and K.R. contributed to scientific discussions.

## Funding

## Competing interests

The authors declare no competing interests.

## Additional information

$^1$Experimentelle Physik 7 and Würzburg-Dresden Cluster of Excellence ctd.qmat, Julius-Maximilians-Universität, Am Hubland, Würzburg, Germany. $^2$Graduate School of Science and Engineering, Chiba University, Inage-ku, Chiba, Japan. $^3$I. Institute for Theoretical Physics and Centre for Free-Electron Laser Science, Universität Hamburg, Hamburg, Germany. $^4$Institut für Experimentalphysik, Universität Hamburg, Hamburg, Germany. $^5$Deutsches Elektronen-Synchrotron DESY, Hamburg, Germany. $^6$The Hamburg Centre for Ultrafast Imaging (CUI), Hamburg, Germany. $^7$Institut für Experimentelle und Angewandte Physik, Christian-Albrechts-Universität zu Kiel, Kiel, Germany. $^8$Institute of Experimental Physics, TU Bergakademie Freiberg, Freiberg, Germany. $^9$Center for Efficient High Temperature Processes and Materials Conversion (ZeHS), Freiberg, Germany. $^{10}$UGC-DAE Consortium for Scientific Research, Indore, Madhya Pradesh, India. $^{11}$European X-Ray Free-Electron Laser Facility, Schenefeld, Germany. $^{12}$Institute of Physics, Brandenburg University of Technology Cottbus-Senftenberg, Cottbus, Germany. $^{13}$Ruprecht Haensel Laboratory, Deutsches Elektronen-Synchrotron DESY, Hamburg, Germany. $^{14}$Kiel Nano, Surface and Interface Science KiNSIS, Kiel University, Kiel, Germany. ✉e-mail: markus.scholz@desy.de

