## [Transparent Peer Review file · Nature Communications]

Femtosecond concerted rotation of molecules on a 2D material interface

Corresponding Author: Dr Markus Scholz

Version 0:

Reviewer comments:

Reviewer #1

(Remarks to the Author)

The manuscript has already undergone a thorough revision in response to the technical comments of one of the referees. The authors have addressed these points carefully and convincingly. In my view, the manuscript is now well-suited for publication in Nature Communications, and I would like to congratulate the authors on this excellent piece of work. I have, however, one technical detail that I would ask the authors to consider before final publication.

In line 121 the authors discuss the change in the carrier relaxation rate in TiSe₂ and how this is influenced by CuPc adsorption. The authors refer to Figure 6 in supplementary materials, which should point to a "significantly prolonged relaxation time". I am not sure about this statement: first of all, I would like to suggest the authors to perform a fit of the curves shown in SI Fig 6a to carefully extract the rising time of the final as well as the decay time, in form of a simple exponential function. Does this fitting procedure really provide a longer relaxation constant for the system with molecules? Judging from the sketch of charge transfer dynamics provided in figure 2f of the main manuscript, I would actually expect a shorter relaxation rate, since the presence of the molecule offers an additional scattering channel (LUMO of CuPs) for the photo-excited carriers in the Ti 3d band, which would actually result in a decrease of the relaxation rate.

Very minor point: in section 1.6 of the supplementary information there is a reference to the supplementary itself that should be removed.

Reviewer #2

(Remarks to the Author)

I appreciate the response of the authors to the questions raised by the reviewers in a previous reviewing round (when their manuscript has been under consideration with Nature Nanotechnology), but unfortunately, I do not think that the authors have addressed the criticism in a sufficient manner, as I will detail below. Therefore, I cannot recommend their manuscript for publication.

(1) In my opinion, their central claim of "unidirectional molecular rotation and transformation to homochiral domains" is insufficiently substantiated. In the Fig.1 of their response, the authors compare the FWHM and $k||$ value of their experimental HOMO ring with simulations claiming that both show the same trend. But in fact, experiment and simulation largely differ in the magnitude of the effect. While in simulation the ring radius decreases by only 0.4%, in experiment 2.5% change in experiment is seen. Even worse, while FWHM in the simulation changes by 1% when going from neutral to the cationic CuPc, experimentally Figure 1 suggests a 30% increase! Given these largely different effects, one cannot claim that one understands the mechanism behind experimentally observed effects.

(2) At many points in the manuscript, in my opinion, the explanations for the observed effects in the data remain too speculative. To give a few examples:

lines 123-124: "this suggests that the adsorbed layer provides additional electron reservoir for the excitation process or

modifies the electron-phonon coupling within TiS₂."

lines 200-203: "While calculations suggest ... molecules rotate in opposite directions ..., our measurements show single-chirality phase, indicating nonreciprocal intermolecular interactions drive the system toward homochirality"

lines 206-207: "... dynamic interplay between molecule-substrate interactions and intermolecular forces" (What does this actually mean?)

(3) The current version of the manuscript still lacks sufficient information about the structural details of the CuPc/TiSe₂ interface. Despite the extensive supporting material, there is nowhere a structural model showing both the substrate with designation of what the authors define as x and y directions, together with overlayer unit cell and a definition of the azimuthal angles with respect to the substrate directions. In my opinion, a clear definition of these structural model is essential to understand the analysis of their experimental and computation data, e.g., to understand how the symmetrization of their data is done. In that respect, an obviously wrong epitaxial superstructure matrix (typo?) as given in line 290, (2, 4.5 / 4, 0.2) is also not helpful. (0.2 ?!)

(4) As another weak point regards the R-factor analysis for the XPD-simulations summarized in Figure S5 of the supplement. First, it is not clear how the 9 different shapes have been chosen? And second, more importantly, the R-factors for the cationic CuPc⁺ shows only little variation over structure numbers #36-43 judging from Fig. S5a. However, these structures correspond to in-plane rotations ranging from 0° to +65°! So, based on this, how can the authors then infer a rotational angle from their experimental XPD data?

Reviewer #3

(Remarks to the Author)

Baumgärtner et al. use the impressive capabilities of HHG-based and FEL-based momentum microscopy to learn about optical excitations and structural deformations in highly interesting organic/inorganic hybrid heterostructures. Moreover, they support their findings by theoretical simulations. The most intriguing capabilities in the experiments thereby is that the authors can simultaneously probe core and valence band states and thus extract a comprehensive information on electronic and structural dynamics.

While the statements and claims of the manuscript are highly exciting, I find them sometimes hard to follow. Hence, at the moment, I find it hard to state whether the major claims of the authors are correct. This includes the clockwise and counterclockwise rotation of CuPc⁰ and CuPc⁺ and also the formation of a homochiral phase. If the statements can be extracted from the data, the publication is suitable for Nature Communications. I would like to ask the authors the following questions for clarification:

- 1) How do the authors know that only 1% of the LUMO are occupied? Are truly single-particle LUMO orbitals occupied or is it an excitonic excitation?
- 2) In Fig. 1c, the authors label Ti bands and the LUMO. They also show a comparison to calculations. Can the authors help me understand what I am seeing in the momentum map? I tried to compare the calculated and the measured map, but they do not seem to be comparable? If the assignment of the authors is correct, experiment and calculation must be comparable.
- 3) In Fig. 1f, major excitation above EF is seen at t₀. Is this related to LAPE, as p-polarized pump pulses are used? In this manner, is also the momentum map in Fig. 1c dominated by LAPE? In how far do the yellow LUMO data points in Fig. 2d agree with the cross-correlation of the laser pulses?
- 4) At many points throughout the manuscript, I would like to ask the authors to do a better job in describing what is plotted, e.g., what are the integration windows? What is the energetic integration width of the momentum maps shown in Fig. 1c-e? This question is especially important, as the authors give energies with an accuracy of 1 meV, while – I guess – they integrate in energy windows >100 meV (?). In the same manner, also the integration width in the momentum maps of Fig. 3a-c should be given.
- 5) Following question 4, how can the authors be sure that the momentum map in Fig. 3c can be assigned only to CuPc⁺ molecules? If I take a look for the fits in Fig. 3c, there will be – depending on the integration width - >10% contribution of CuPc⁰ molecules.
- 6) The calculated and measured HOMO map in Fig. 1e at t₋₁ indeed seem comparable. For larger delays, it seems like the 'blob' structure is washed out. However, I guess this is the same data used in Fig. 3a for the identification of the \alpha, \beta, \gamma features?
- 7) Based on the data in Fig. S6b, the authors discuss about a 'significant' change in the dynamics. Can the authors put this argument on a more quantitative level than only the visual comparison of the black and the other colored lines? After fitting, is the proposed effect of hole transfer larger than the error margins?
- 8) How are the integration windows chosen for the data evaluation shown in Fig. S6b? Might pump-induced broadening of the HOMO and the 2D material bands impact the observation?
- 9) Can the authors also address Ti or Se core-levels? This would allow a direct comparison of the dynamics with and without CuPc adsorbed.
- 10) In Fig. 2c, why is the 288 eV peak not found in theory (at delay t₋₁)? Is this peak related to the 287 eV peak at t₁, which is assigned to CuPc⁺?
- 11) In Fig. 2c (delay t₁), for energies >-286 eV, 3 peaks are used for fitting (1 red, 2 blue). I guess the fit would also be reasonable with only two peaks? How good is the statement, that CuPc⁺ molecules contribute?
- 12) Following on 11, how good is the statement that ~50% of all molecules are charged? What are the arguments for this? The different dynamics in Fig. 6b and the fitting in Fig. 2c?

13) The direct comparison of the t-1 and t1 insets in Fig. 3a with regard to the α , β , γ features is very exciting. Are the new β and γ features visible for all azimuthal angles? What is the momentum-width for the line-profiles shown in the main panel?

14) Following on 13, what is the delay-dependence of the azimuthal offset of the β and γ features?

15) Considering only the trOT data in Fig. 3a, how do the authors know that the differently charged molecules are rotated by $+15^\circ$ and/or -15° , respectively? How can this statement be based on the trOT data?

16) How are the line-profiles in Fig. 3b obtained? Momentum-width? I tried to visually compare the calculated and measured momentum maps, but I failed. Can the authors help me here?

17) If the CuPc0 and CuPc+ molecules are rotating in opposite direction, this should be visible in the trXPD pattern at $\sim 284\text{eV}$ and $\sim 285\text{eV}$. While the line-profile analysis in Fig. 3b for CuPc0 is nice, why don't the authors show the same for the 285eV data, i.e., for CuPc+? This might be a very strong experimental point...

18) Hence, I cannot follow the argument of the authors that CuPc+ and CuPc0 are rotating in opposite direction.

19) From experiment, can the authors explain how they conclude on the formation of a homochiral film?

Version 1:

Reviewer comments:

Reviewer #1

(Remarks to the Author)

My concerns have been successfully addressed in the revision and I think that the manuscript is now ready for publication.

Reviewer #2

(Remarks to the Author)

I appreciate the authors' efforts in clarifying the points, which have been raised by the Reviewers in the previous round. This has certainly helped to improve their manuscript, however, I have still some reservations that also Reviewer 3 seems to share. Since my main concern regards one central aspect of their paper, namely the experimental evidence for opposite rotation of neutral and charged molecules and the formation of homochiral domains, I kindly ask the authors to clarify their reasoning for the interpretation of the data.

Let me try to summarize my understanding of the authors' experimental findings and reasoning regarding the rotation of the molecules, which I base on the information, which is currently, scattered in the main part and Supporting Information as well as in their rebuttal letter:

(i) The trOT data shows a rotation of $+15^\circ$ and -15° at t1, but provide no directional information, and the trOT data does not allow to distinguish between neutral and charged molecules.

(ii) From inspection of Fig. S6, I deduce that the R-factor analysis of the trXPD data at t1 for the neutral CuPc indicates a minimum at -15° (structure #25), while the structure at $+15^\circ$ (structure #19) has a significantly worse R-factor.

(iii) For charged molecules, the R-factor analysis of the trXPD data does not yield an unambiguous rotational angle within the error margin. (structures #36-43 are actually all for rotations in + direction, why are no structures in the "-" direction considered?)

(iv) The trXPS data suggests a ratio of charged:neutral molecules to be 45%.

(v) Steric constraints in a full monolayer CuPc film, the rotation of neutral molecules by -15° , see point (ii), implies a correlated motion of neighboring molecules in the opposite direction to avoid collisions.

If my above summary is correct, then I kindly ask the authors to clearly state their line of arguments based on experimental findings. In the current version of the manuscript, their claim "observation of femto-second unidirectional rotation stabilizing a homochiral domain" appears to be repeated three times in the abstract, Introduction and Conclusion, respectively, but in the results section, it is difficult to follow the authors' reasoning leading them to the interpretation of their data.

Reviewer #3

(Remarks to the Author)

I would like to thank the authors for their reply and the update to the manuscript. The manuscript has improved in terms of clarity, even though at some points I still find it to be not as direct as it could be. Here are two examples, but there are more of such points throughout the text:

• Page 4, line 164/165: Why don't the authors label the three and the six peaks in Fig. 3b? It is not straightforward to find 3 and 6 peaks in the figure.

• Page 4, line 172,173: It might be common knowledge that the γ point is at normal emission, but it would still be helpful if the authors could be more precise when describing the difference between γ , δ , and ϵ .

Even though there might be still room for improvement of the manuscript presentation (as exemplified above), the new passages of the manuscript now explain what is obtained from data and what is obtained from interpretation. I would suggest that the authors also update the abstract and the introduction in this regard. For example:

• "... we uncover the synchronized rotational motion of molecules on..." This is an interpretation of the data, right? Even though the authors report a time-resolved measurement, the synchronous motion is not directly measured.

• "... and the formation of homochiral domains." Also, this is an interpretation following static LEED and tr experiment

analysis.

Nevertheless, I would like to congratulate the authors on their findings. I believe the updated manuscript is now suitable for publication. (Still, I encourage the authors on further improving the presentation of their great results).

REVIEWER COMMENTS

Reviewer #1 (Remarks to the Author):

The manuscript has already undergone a thorough revision in response to the technical comments of one of the referees. The authors have addressed these points carefully and convincingly. In my view, the manuscript is now well-suited for publication in Nature Communications, and I would like to congratulate the authors on this excellent piece of work.

I have, however, one technical detail that I would ask the authors to consider before final publication.

In line 121 the authors discuss the change in the carrier relaxation rate in TiSe₂ and how this is influenced by CuPc adsorption. The authors refer to Figure 6 in supplementary materials, which should point to a "significantly prolonged relaxation time". I am not sure about this statement: first of all, I would like to suggest the authors to perform a fit of the curves shown in SI Fig 6a to carefully extract the rising time of the final as well as the decay time, in form of a simple exponential function. Does this fitting procedure really provide a longer relaxation constant for the system with molecules? Judging from the sketch of charge transfer dynamics provided in figure 2f of the main manuscript, I would actually expect a shorter relaxation rate, since the presence of the molecule offers an additional scattering channel (LUMO of CuPs) for the photo-excited carriers in the Ti 3d band, which would actually result in a decrease of the relaxation rate.

We thank the reviewer for this helpful suggestion. Following the recommendation, we fitted the population dynamics in Fig. S6 using a single-exponential function. For 1 ML CuPc on TiSe₂ the extracted decay time is $\tau_{decay} = (0.441 \pm 0.021)$ ps, which is slightly longer than for pristine TiSe₂ (0.386 ± 0.054) ps. However, the difference lies within the combined 1 σ uncertainty.

Instead, we interpreted intensity maximum t_{max} as the more robust indicator: the conduction-band population peaks at $t_{max} = (0.369 \pm 0.041)$ ps for CuPc/TiSe₂ and at $t_{max} = (0.181 \pm 0.054)$ ps for pristine TiSe₂ at similar fluence (see Fig. R2 and Table 1). At lower fluence (≈ 0.30 mJ cm⁻²) both systems behave identically within the error.

To clarify this point in the manuscript, we now explicitly distinguish between the rise *time*, the decay *constant* and the delay of the population maximum t_{max} . The latter shifts to longer delays upon CuPc adsorption, consistent with an additional transient reservoir for excited electrons.

The microscopic mechanism behind this delay can be rationalized by impact-ionization–induced secondary carriers, as discussed by [1, 2, 3]. Upon photoexcitation, primary hot electrons in TiSe₂ relax through Auger-like scattering, generating secondary carriers whose lifetime depends on energy and carrier density. The presence of CuPc introduces an additional electronic reservoir via the HOMO, which may prolong the cascade by slightly delaying the maximum of the conduction-band population rather than by altering the intrinsic decay constant. We stress that this mechanism is not required for our main conclusions and merely offers a microscopic rationale for the observed delay.

These aspects are now clarified in the revised manuscript like follows:

Line 121ff: "Experimentally, we find that the conduction-band population of the TiSe₂ reaches its maximum intensity latter for CuPc/TiSe₂ ($t_{max} = 0.369 \pm 0.041$ ps) than for pristine TiSe₂ ($t_{max} = 0.181 \pm 0.054$ ps). The corresponding decay constants are $\tau = 0.441 \pm 0.021$ ps and $\tau = 0.386 \pm 0.054$ ps, respectively, and slightly longer for CuPc-covered surface but within the combined uncertainty (Fig. S6 b, Supplementary Materials). This shift of tmax to longer delays indicate that the excited-state population evolves through an intermediate stage before relaxation. We attribute this behavior to an extended cascade of cascade of secondary hot carriers generated by inverse Auger (impact-ionization) processes (7,8), which are further prolonged by charge exchange with the CuPc layer. This observation highlights the

critical role that the molecular environment plays in tuning the dynamics of 2D materials, effectively creating a hybrid system with modified electronic properties."

Fig. R1. **Excitation mechanisms of hot carriers.** A) Direct inter-band excitation generates hot electrons (black circle) and hot holes (white circle) through photon absorption (green arrow). B) Indirect intraband excitation combines photon absorption with a scattering event (dashed black arrow) that alters the hot electron momentum. Processes A) and B) create primary hot carriers, while secondary hot carriers arise from C) Auger scattering and D) impact ionization. Adapted from [1, 2, 3].

We also added the following table to the supplemental materials.

Table 1. Extracted rising and decay constants as well as maximum of intensity for pristine TiSe₂ and CuPc/TiSe₂ samples.

	fluence	τ_{rising}	τ_{decay}	t_{max}
TiSe ₂ + 1 ML CuPc	0.64 mJ/cm ²	(0.065 ± 0.013) ps	(0.443 ± 0.021) ps	(0.369 ± 0.041) ps
Pristine TiSe ₂	0.64 mJ/cm ²	(0.190 ± 0.158) ps	(0.386 ± 0.054) ps	(0.181 ± 0.054) ps
Pristine TiSe ₂	0.30 mJ/cm ²	(0.109 ± 0.153) ps	(0.340 ± 0.022) ps	(0.251 ± 0.013) ps
TiSe ₂ + 0.2 ML CuPc	0.27 mJ/cm ²	(0.083 ± 0.051) ps	(0.393 ± 0.031) ps	(0.197 ± 0.028) ps

Fig. R2. Population dynamics of pristine TiSe₂ and CuPc/TiSe₂ samples as well as fitted rising and decay functions. In addition is shown a gaussian peak with a FWHM of 95 fs at $t_0=0$.

[1] H. Petek and S. Ogawa, "Femtosecond time-resolved two-photon photoemission studies of electron dynamics in metals", Progress in Surface Science 56, 239–310 (1997).

[2] S. Mathias, S. Eich, J. Urbancic, S. Michael, A. V. Carr, S. Emmerich, A. Stange, T. Popmintchev, T. Rohwer, M. Wiesenmayer, A. Ruffing, S. Jakobs, S. Hellmann, P. Matyba, C. Chen, L. Kipp, M. Bauer, H. C. Kapteyn, H. C. Schneider, K. Rossnagel, M. M. Murnane, and M. Aeschlimann, "Self-amplified photo-induced gap quenching in a correlated electron material", *Nature Communications* 7, 12902 (2016).

[3] M. Aeschlimann, M. Bauer, and S. Pawlik, "Competing nonradiative channels for hot electron induced surface photochemistry", *Chemical Physics* 205, 127–141 (1996).

Very minor point: in section 1.6 of the supplementary information there is a reference to the supplementary itself that should be removed.

Thank you very much for that carefully reading the supplementary! We removed that reference but added also the sample of the fitted rising and decay functions.

Reviewer #2 (Remarks to the Author):

I appreciate the response of the authors to the questions raised by the reviewers in a previous reviewing round (when their manuscript has been under consideration with *Nature Nanotechnology*), but unfortunately, I do not think that the authors have addressed the criticism in a sufficient manner, as I will detail below. Therefore, I cannot recommend their manuscript for publication.

(1) In my opinion, their central claim of "unidirectional molecular rotation and transformation to homochiral domains" is insufficiently substantiated. In the Fig.1 of their response, the authors compare the FWHM and $k_{||}$ value of their experimental HOMO ring with simulations claiming that both show the same trend. But in fact, experiment and simulation largely differ in the magnitude of the effect. While in simulation the ring radius decreases by only 0.4%, in experiment 2.5% change in experiment is seen. Even worse, while FWHM in the simulation changes by 1% when going from neutral to the cationic CuPc, experimentally Figure 1 suggests a 30% increase! Given these largely different effects, one cannot claim that one understands the mechanism behind experimentally observed effects.

We thank the reviewer for raising this important point again. There seems to be a misunderstanding: the rotation of the HOMO is deduced from the appearance of the additional features beta and gamma at t_1 (Fig. 3 a) and not from the change of $k_{||}$ of the HOMO ring.

The signature of the HOMO is a ring like structure with 12 bright intensities distributed around the ring, see Fig. 1 e and Fig. S9. The signature of the rotation in the orbital tomography data thus appears as two additional features ± 15 degree next to each of original peaks and shown in Fig. 3 a.

We added to our manuscript:

In line 156ff: "As shown in Fig. 3 a, the HOMO electron density distribution forms a ring-like structure with 12 distinct peaks (α) in momentum space. At t_1 , we observe additional features (β and γ) separated from the initial peaks by about $-15^\circ \pm 3^\circ$ and $+15^\circ \pm 3^\circ$, respectively, indicating rotational motion of the molecules. The trOT data shows the occurrence and magnitude of the molecular rotation but do not provide directional information of the charged and neutral molecules."

(2) At many points in the manuscript, in my opinion, the explanations for the observed effects in the data remain too speculative. To give a few examples:

We have refocused the discussion of our manuscript and separated the experimental observations and interpretation more carefully.

lines 123-124: "this suggests that the adsorbed layer provides additional electron reservoir for the excitation process or modifies the electron-phonon coupling within TiS2."

We observe that the depletion of the HOMO closely follows the increase of the Ti 3d states, with the minima and maxima of the integrated intensities occurring at the same delay (see Fig. 2 d). In addition, we observe a transient binding energy shift of the C 1s core level. Our C 1s calculation show, that the core level shift results from an electron transfer between the substrate and the electron (see Fig. 2 a, c, d). We explain the combined valence

band, HOMO and core level dynamics as signature of charge exchange between the molecules and the substrate. This is illustrated in Fig. 2 d of the manuscript.

lines 200-203: "While calculations suggest ... molecules rotate in opposite directions ..., our measurements show single-chirality phase, indicating nonreciprocal intermolecular interactions drive the system toward homochirality"

Our pair-potential calculations, which model the orientation of molecules based on Coulomb and van der Waals interactions between neutral and charged species, do not include nonreciprocal forces. We therefore offered this as a possible explanation rather than a firm conclusion. To prevent any misunderstanding, we revised our manuscript and offer the formation of homochiral domains as a logical consequence of the missing clockwise rotation of charged molecules.

Line 217ff: "The absence of opposite-rotation features therefore implies that a homochiral molecular configuration emerges under photoexcitation. This behavior is consistent with other energy-driven molecular systems on surfaces (33,34,35,36). The homochiral arrangement vanishes once the excitation is removed, confirming that the effect is reversible and not due to permanent structural modification."

lines 206-207: "... dynamic interplay between molecule-substrate interactions and intermolecular forces" (What does this actually mean?)

Both, the molecule-substrate interaction and the intermolecular forces, act simultaneously: on the one hand, charge transfer between molecule and substrate modifies the potential landscape and on the other hand, Coulomb interactions between neutral and charged molecules reshape the intermolecular forces. Our observations cannot be explained by only one of these effects. Rather, both mechanisms are relevant, as supported by the dielectric model and pair-potential calculations.

(3) The current version of the manuscript still lacks sufficient information about the structural details of the CuPc/TiSe₂ interface. Despite the extensive supporting material, there is nowhere a structural model showing both the substrate with designation of what the authors define as x and y directions, together with overlayer unit cell and a definition of the azimuthal angles with respect to the substrate directions. In my opinion, a clear definition of these structural model is essential to understand the analysis of their experimental and computation data, e.g., to understand how the symmetrization of their data is done. In that respect, an obviously wrong epitaxial superstructure matrix (typo?) as given in line 290, (2, 4.5 / 4, 0.2) is also not helpful. (0.2 ?!)

Our superstructure matrix is correct and deduced from LEED measurements in Fig. S1 of the supplemental materials. The structural arrangement is presented in Fig. 3. f. The LEED and from it deduced structural arrangement is shown in addition shown in Fig. R3.

Fig. R3. Structural characterization of CuPc/TiSe₂. A LEED image of a monolayer CuPc atop TiSe₂ taken at 24 eV. Unit cell vectors of the molecule and the substrate are marked by red and blue arrows, respectively. B Schematic representation of molecules adsorbing along the substrate's high-symmetry directions as in point-on-line growth. C Molecular arrangement within one growth domain.

At a coverage of one monolayer, the CuPc monolayer on TiSe₂ forms a so-called point-on-line structure. In this epitaxial arrangement, the overlayer is commensurate with the substrate along one crystallographic direction,

while in the perpendicular direction the registry is incommensurate. This is expressed by the superstructure matrix (2, 4.5 / 4, 0.2) as stated in our manuscript, which means that along the TiSe₂ [100] direction the molecular lattice coincides exactly with every second substrate lattice vector (the "point-on" condition). Along the perpendicular direction, however, the molecular rows do not match a simple integer ratio, but instead form a quasi-incommensurate periodicity.

To avoid any misunderstanding, we added:

Line 294ff: "This epitaxial relationship corresponds to a point-on-line superstructure: the CuPc overlayer is commensurate with the TiSe₂ substrate along one crystallographic direction, while the perpendicular direction exhibits a quasi-incommensurate registry. From this superstructure, a real-space arrangement of densely packed flat-lying CuPc molecules with an intermolecular spacing of 13.8 Å can be determined."

(4) As another weak point regards the R-factor analysis for the XPD-simulations summarized in Figure S5 of the supplement. First, it is not clear how the 9 different shapes have been chosen? And second, more importantly, the R-factors for the cationic CuPc⁺ shows only little variation over structure numbers #36-43 judging from Fig. S5a. However, these structures correspond to in-plane rotations ranging from 0° to +65°! So, based on this, how can the authors then infer a rotational angle from their experimental XPD data?

We extracted the molecular rotation of neutral and charged species using two complementary methods: time-resolved orbital tomography and time-resolved photoelectron diffraction.

The trXPD for neutral CuPc, the R-factor analysis of the XPD data indicates a minimum around +15°. For CuPc⁺ at t_1 , the R-factor varies only weakly across #36-43 ($\Delta R < 0.02$), indicating that the XPD contrast is dominated by molecular deformation upon charging rather than by the azimuthal rotation itself. Therefore, XPD alone cannot provide an unambiguous rotation angle for the cationic species.

The orbital tomography data, in contrast, provide a clearer indication of rotations of approximately -15° and +15°. Considering steric constraints in the densely packed film, a concerted motion of both species is the only self-consistent interpretation.

We now make this reasoning explicit in the revised text and discuss the respective sensitivity limits of both techniques. We revised the manuscript:

Line 175ff: "For charged molecules, the R-factor analysis does not yield an unambiguous rotational angle within the error margin. However, steric constraints must also be considered: at a coverage of one monolayer, the molecules are densely packed. In such an arrangement, a rotation of neutral molecules necessarily implies correlated motion of neighboring molecules to avoid collisions. The XPD pattern and calculation for CuPc+t1 molecules rotated CCW by about +15° ± 3° is shown in Fig. 3 c."

Line 209ff: "The trOT quantifies magnitude of the azimuthal rotation, while the trXPD data provide its direction."

Line 192ff: "This ultrafast rotation is captured in both tomographic trOT snapshots of the HOMO (Fig. 3 a) and trXPD patterns of the C 1s core level (Fig. 3 b, c) within ~375 fs after absorption of the pump pulse."

Reviewer #3 (Remarks to the Author):

Baumgärtner et al. use the impressive capabilities of HHG-based and FEL-based momentum microscopy to learn about optical excitations and structural deformations in highly interesting organic/inorganic hybrid heterostructures. Moreover, they support their findings by theoretical simulations. The most intriguing capabilities in the experiments thereby is that the authors can simultaneously probe core and valence band states and thus extract a comprehensive information on electronic and structural dynamics.

While the statements and claims of the manuscript are highly exciting, I find them sometimes hard to follow. Hence, at the moment, I find it hard to state whether the major claims of the authors are correct. This includes the clockwise and counterclockwise rotation of CuPc⁰ and CuPc⁺ and also the formation of a homochiral phase. If the statements can be extracted from the data, the publication is suitable for **Nature Communications**. I would like to ask the authors the following questions for clarification:

1) How do the authors know that only 1% of the LUMO are occupied? Are truly single-particle LUMO orbitals occupied or is it an excitonic excitation?

We thank the reviewer for these constructive and detailed questions. The previously stated ~1% LUMO occupation was derived from the transient intensity ratio between the LUMO signal and the total molecular contribution in the trARPES maps. Because the pump is off-resonant with the HOMO–LUMO gap, direct optical excitation into the LUMO is inefficient, and the weak signal observed is consistent with hot-electron scattering from the Ti 3*d* band.

Since this estimate is not essential for our discussion of charge transfer and molecular charging, we have removed the numerical value from the revised manuscript and now simply state that a minor transient occupation of the LUMO is observed. The excitation involves delocalized scattering electrons rather than a bound exciton state, consistent with the short (< 100 fs) decay constant of the LUMO signal (see Fig. R4).

From our many-body calculations of the ground and excited states of CuPc, we can conclude that excitonic effects for the excited state involving LUMO are negligible. The excited state can be very well approximated by a single-particle picture. Namely, we use the configuration interaction method. In this method, the excited states are represented as a linear combination of configuration state functions. We can see that the contribution to the first excited state is dominated by a configuration with a singly occupied HOMO and a singly occupied LUMO as written in the supplementary information, Section 2.3.4:

“The dominant contribution to the first excited state is given by a configuration with a singly occupied HOMO and a singly occupied LUMO, and the dominant contribution of the second excited state is given by a configuration with a singly occupied HOMO and a singly occupied LUMO’ orbitals.”

2) In Fig. 1c, the authors label Ti bands and the LUMO. They also show a comparison to calculations. Can the authors help me understand what I am seeing in the momentum map? I tried to compare the calculated and the measured map, but they do not seem to be comparable? If the assignment of the authors is correct, experiment and calculation must be comparable.

Thanks for that question. We would like to highlight the calculation in Fig. S11 of the Supplemental Material. There, the intensity at Γ is associated with the Ti 3*d* bands, while the calculated LUMO and LUMO’ intensities are shown in Section 2.3.4 of the Supplemental Material. Importantly, these orbitals exhibit no intensity at Γ .

3) In Fig. 1f, major excitation above EF is seen at t_0 . Is this related to LAPE, as p-polarized pump pulses are used? In this manner, is also the momentum map in Fig. 1c dominated by LAPE? In how far do the yellow LUMO data points in Fig. 2d agree with the cross-correlation of the laser pulses?

That is an excellent question. Yes, the sample was pumped with 1.55 eV *p*-polarized light.

- For LAPE we would expect a LAPE replica from the Fermi edge, ~1.55 eV above EF (see Fig. R1).
- Likewise, the strong signal from the HOMO, centered at –310 meV below EF, would produce a strong LAPE signal around 1.24 eV above EF if present. Neither feature is observed.
- In addition, the momentum distribution of the transient signal differs markedly from that expected for a LAPE replica: the map at 180 meV above EF exhibits the threefold symmetry characteristic of the Ti 3*d* conduction bands, rather than the momentum pattern of the Se 4*p* valence bands at –1.37 eV.
- Moreover, the extracted LUMO decay constant of 94 fs is comparable to the instrument response function, indicating an instrument-limited upper bound of the true decay time.

Taken together, the absence of spectral replicas, the distinct *k*-space signature, and the fluence/polarization behavior all confirm that the observed feature is a genuine ultrafast population decay, not a LAPE artifact.

Fig. R4. Upper part) Iso-energy cut with at time zero with an energy integration window of 200 meV around the energy given in the respective picture. Lower part) Intensity population of LUMO with $\tau = 94$ fs and Gaussian pump pulse with a FWHM of 95 fs.

We added this analysis to the supplemental materials in section 1.5.

4) At many points throughout the manuscript, I would like to ask the authors to do a better job in describing what is plotted, e.g., what are the integration windows? What is the energetic integration width of the momentum maps shown in Fig. 1c-e? This question is especially important, as the authors give energies with an accuracy of 1meV, while – I guess – they integrate in energy windows $>100\text{meV}$ (?). In the same manner, also the integration width in the momentum maps of Fig. 3a-c should be given.

All shown momentum maps in Fig. 1 c-e are integrated over ± 220 meV around the denoted center energy. The integration range of Fig. 3 a is the same as in Fig. 1 e for $t-1$ and $t1$. We will add this information in Fig. 1 and 3. The integration range in Fig. 3 c is showed in Fig. R2 for the corresponding charged state and time delay.

These values are now added to the figure captions:

Fig. 1. "For (c)-(e) we integrated in energy over ± 220 meV around the denoted center energy."

5) Following question 4, how can the authors be sure that the momentum map in Fig. 3c can be assigned only to CuPC+ molecules? If I take a look for the fits in Fig. 3c, there will be – depending on the integration width - $> 10\%$ contribution of CuPc0 molecules.

We extracted the XPD pattern from specific C 1s energy windows (Fig R2) with marked integration ranges for XPD patterns of the carbon atoms in CuPc (purple), carbon atoms in CuPc+ (green) and the carbon atoms in CuPc0 (pink). In particular for $t1$ we selected the ranges such, to avoid too much overlap between CuPc+ and CuPc0.

Fig. R5. C 1s core level spectra at $t-1$ and $t1$ are with marked integration ranges for XPD patterns of B the carbon atoms in CuPc (purple), carbon atoms in CuPc+ (green) and D the carbon atoms in CuPc0 (pink). Fits including gaussian shaped peaks for carbon atoms in different chemical environments (solid lines, peaks 1-5) and for shake-up satellites (dashed lines) are included to the spectra at $t-1$ and $t1$.

6) The calculated and measured HOMO map in Fig. 1e at $t-1$ indeed seem comparable. For larger delays, it seems like the 'blob' structure is washed out. However, I guess this is the same data used in Fig. 3a for the identification of the α , β , γ features?

Yes. it is the same sample but taken at a fixed delay time at $t1$. We decided to show in Fig. 1. data extracted from only one data set. In Fig 3 a, we collected much more data at a fixed delay at $t1$.

7) Based on the data in Fig. S6b, the authors discuss about a 'significant' change in the dynamics. Can the authors put this argument on a more quantitative level than only the visual comparison of the black and the other colored lines? After fitting, is the proposed effect of hole transfer larger than the error margins?

There were two aspects we evaluated: first, the maximum transient population above EF and the decay time of the population. The decay times are summarized in Table 1. We have fitted the intensity evolution for Fig. S6 b and show it in the reply of referee #1.

We added to the manuscript:

Line 121ff: "Experimentally, we find that the conduction-band population of the TiSe₂ reaches its maximum intensity latter for CuPc/TiSe₂ ($t_{max} = 0.369 \pm 0.041$ ps) than for pristine TiSe₂ ($t_{max} = 0.181 \pm 0.054$ ps). The corresponding decay constants are $\tau = 0.441 \pm 0.021$ ps and $\tau = 0.386 \pm 0.054$ ps, respectively, and slightly longer for CuPc-covered surface but within the combined uncertainty (Fig. S6 b, Supplementary Materials)."

8) How are the integration windows chosen for the data evaluation shown in Fig. S6b? Might pump-induced broadening of the HOMO and the 2D material bands impact the observation?

We integrated all intensity above the Fermi level originating from the Ti 3d conduction band. For the HOMO intensity evolution shown in Fig. 2 d, the integration window was adjusted to account for both the broadening and binding-energy shift of the HOMO, as illustrated in Fig. S4 a, b.

9) Can the authors also address Ti or Se core-levels? This would allow a direct comparison of the dynamics with and without CuPc adsorbed.

The Se 3p core levels for a coverage of 1 ML is shown in Fig. 2. b, and the Ti 3p in Fig R5. The energy shift is about about 200 meV. The Ti 3p core level also for 1 ML coverage shows a similar shift in binding energy.

Fig. R6. Pump probe scan of $\text{TiSe}_2/\text{CuPc}$. Ti 3p core level evolution.

10) In Fig. 2c, why is the 288eV peak not found in theory (at delay t_1)? Is this peak related to the 287eV peak at t_1 , which is assigned to CuPc^+ ?

At around -288 eV are well known shake-up satellites, which are not included in the calculations. Those are highlighted in Fig. R7 by the dotted lines.

11) In Fig. 2c (delay t_1), for energies $> -286\text{eV}$, 3 peaks are used for fitting (1red, 2 blue). I guess the fit would also be reasonable with only two peaks? How good is the statement, that CuPc^+ molecules contribute?

In addition, we fitted the spectra also according to literature and shown in Fig R6. We selected the integration window such, to minimize the contribution of the CuPc^0 molecules t_1 .

Fig. R7. Pump probe scan of $\text{CuPc}/\text{TiSe}_2$. A) C 1s scan. B) Selected EDC at times indicated in A). C) Carbon atoms for CuPc^0 CuPc^+ and CuPc before and after t_0 .

12) Following on 11, how good is the statement that ~50% of all molecules are charged? What are the arguments for this? The different dynamics in Fig. 6b and the fitting in Fig. 2c?

The energetic separation between chemically inequivalent carbon species in the C 1s spectra allows us to disentangle the signatures of CuPc⁺ and CuPc⁰. We analyzed the trXPS peak ratios at $t-1$ and $t1$. By comparison of relative peak areas we quantify the maximal fraction of positively charged molecules CuPc⁺/CuPc to be $(45\pm 2)\%$. Here, we explored two ways for determining the intensity ratio of the neutral molecules before time zero and the signature of the neutral molecules at $t1$. First by the intensity of all peaks and second by evaluating only peak 1 at $t-1$ and $t1$. Both show a similar result.

13) The direct comparison of the $t-1$ and $t1$ insets in Fig. 3a with regard to the α , β , γ features is very exciting. Are the new β and γ features visible for all azimuthal angles? What is the momentum-width for the line-profiles shown in the main panel?

Yes, it is visible at different angles but at some positions intermixed with the intensity at M - and second gamma points. Fig. R7 shows the beta and gamma feature at different azimuthal angle. The integration range was about 0.1 \AA^{-1} .

Fig. R8. Azimuthal line cuts of HOMO at $t-1$ and $t1$.

We added to the Figures:

Line 612: Fig. 1. "For (c)-(e) we integrated in energy over $\pm 220 \text{ meV}$ around the denoted center energy."

14) Following on 13, what is the delay-dependence of the azimuthal offset of the β and γ features?

Unfortunately, during the beamtime the repetition rate of current FELs is too low to extract XPD pattern within a temporal resolution of 100 fs. For this particular system we had to integrate within a time window of 375 fs after time zero.

15) Considering only the trOT data in Fig. 3a, how do the authors know that the differently charged molecules are rotated by $+15^\circ$ and/or -15° , respectively? How can this be statement be based on the trOT data?

We can see that the γ feature shifts slightly towards gamma, see Fig. 3 a, inlay at $t1$. Our calculations show that this might be related to the charged molecule. But our calculations are limited and thus we decided not to include that aspect in the current manuscript.

16) How are the line-profiles in Fig. 3b obtained? Momentum-width? I tried to visually compare the calculated and measured momentum maps, but I failed. Can the authors help me here?

The line plots are extracted from the inlayed pictures on the right. This is just a different projection in the azimuth and polar projection from the inlay on the right. Line profiles were obtained at 0.7 \AA^{-1} by averaging over a $\Delta k = \pm 0.15 \text{ \AA}^{-1}$ window around the azimuthal cuts. We added to the figure caption:

Line 644ff: Fig. 3. “The azimuthal line cuts were extracted at $(0.7 \pm 0.15) \text{ \AA}^{-1}$ between $(282.6 \text{ eV} - 284.1 \text{ eV})$ binding energy at t_{-1} and between $(284.1 - 284.8) \text{ eV}$ and $(282.7 - 283.6) \text{ eV}$ binding energy for neutral and charged molecules at t_1 .”

And in the manuscript:

Line 165ff: “Figure 3 b shows the trXPD pattern and the extracted neutral molecule contributions. At t_{-1} , the neutral molecule signal is integrated between -282.6 eV and -284.1 eV . At t_1 , the neutral and charged molecules are isolated by integrating the C 1s signal between -284.1 eV and -284.8 eV (neutral) and between -282.7 eV and -283.6 eV (charged), respectively.”

17) If the CuPc0 and CuPc+ molecules are rotating in opposite direction, this should be visible in the trXPD pattern at $\sim 284\text{eV}$ and $\sim 285\text{eV}$. While the line-profile analysis in Fig. 3b for CuPc0 is nice, why don't the authors show the same for the 285eV data, i.e., for CuPc+? This might be a very strong experimental point...

Figure R6 displays line cuts of the charged (CuPc0, green line, peak 4 at t_1) and neutral (CuPc0, t_{-1}) species. The XPD pattern of CuPc+ differs markedly from that of the neutral molecules, indicating structural changes upon charging. To analyze these differences, we calculated corresponding XPD patterns and performed an R -factor analysis. The comparison shows that the charged molecules are distorted in a manner consistent with the deformation illustrated in Fig. 3 b. However, the rotation component cannot be unambiguously extracted from this analysis alone, as it is partly convoluted with the molecular deformation. Nonetheless, we can also identify non-rotated molecular contributions in the experimental data, similar to the line-cut behavior observed for neutral molecules at t_1 . We will clarify this point in the revised manuscript and explicitly refer to Fig. R6 for the charged-state analysis.

Fig. R9. EDC at t_{-1} and t_1 , XPD pattern of CuPc+ and azimuthal line cut with a width of 0.4 \AA^{-1} .

18) Hence, I cannot follow the argument of the authors that CuPc+ and CuPc0 are rotating in opposite direction.

The rotation of CuPc0 is directly extracted from the experimental XPD patterns and confirmed by an R -factor analysis. For CuPc+, the R -factor analysis does not yield an unambiguous rotational angle within the error margin. However, steric constraints must also be considered: at a coverage of approximately one monolayer, as determined from LEED measurements prior to time zero (Fig. 3 f), the molecules are densely packed. In such an arrangement, a rotation of CuPc0 necessarily implies correlated motion of neighboring molecules to avoid collisions. This geometric coupling requires a dominant mixed arrangement of CuPc0 and CuPc+ species, as illustrated in Fig. 3 g. An alternative scenario of separated neutral and charged patches would not provide a driving force for rotation within the neutral domains. Only in a mixed configuration can the observed motion be explained by Coulomb interactions between neighboring molecules.

We rewrote this section in the manuscript:

Line 171ff: “In an azimuthal line extracted at $(0.7 \pm 0.15) \text{ \AA}^{-1}$ from the trXPD data at t_{-1} and t_1 (Fig. 3 b) shows that the δ and ϵ features of the neutral molecules are rotated clockwise by $-15^\circ \pm 3^\circ$. In addition, the δ' and ϵ' appear shifted toward the Γ point for CuPc0 t_1 , indicating a decreased adsorption height. The XPD pattern of charged molecules at t_1 differs markedly from that at t_{-1} , indicating that the molecule undergoes a structural deformation. Multiple scattering calculations for different model geometries reveal out-of-plane

deformation with benzene wings approaching the substrate (Fig. 3 d). For charged molecules, the R-factor analysis does not yield an unambiguous rotational angle within the error margin. However, steric constraints must also be considered: at a coverage of one monolayer, the molecules are densely packed. In such an arrangement, a rotation of neutral molecules necessarily implies correlated motion of neighboring molecules to avoid collisions. The XPD pattern and calculation for CuPc+t1 molecules rotated CCW by about $+15^{\circ} \pm 3^{\circ}$ is shown in Fig. 3 c."

19) From experiment, can the authors explain how they conclude on the formation of a homochiral film?

Static LEED measurements prior to time zero revealed the presence of mirror domains in the as-prepared CuPc/TiSe₂ monolayer. In such mirror domains, any azimuthal rotation of the molecules would be expected to occur in opposite directions, as also indicated by our pair-potential calculations. However, in the time-resolved XPD data we do not observe signatures of opposite rotations. Instead, the transient diffraction patterns are consistent with a uniform, unidirectional rotation of all molecules within the probed area. Based on this absence of counter-rotation signals, we infer that a homochiral film develops during excitation, similar to reports for other driven molecular systems on surfaces.

In our revised manuscript, we provide a clearer distinction between experimental results and their interpretation. The formation of homochiral domains is proposed as a possible explanation of the uniform rotation of charged and neutral molecules extracted from XPD measurements.

Reviewer #1

My concerns have been successfully addressed in the revision and I think that the manuscript is now ready for publication.

We thank Reviewer #1 for the positive assessment!

Reviewer #2

We thank Reviewer #2 for the detailed and careful summary of our experimental findings and for clearly articulating the remaining concerns.

We confirm that the Reviewer's summary is essentially correct. This logical structure is now explicitly stated and followed in the Results section of the revised manuscript. To address the Reviewer's concern that the reasoning was previously difficult to follow, we have revised the Results section to make the following step-by-step argument explicit. No new interpretation is introduced; rather, the revised text makes explicit the experimental logic already underlying the analysis:

Line 162ff "While the trOT data shows the occurrence and magnitude of the molecular rotation, it does not provide directional information or distinguish between charged and neutral molecules.

Similar rotations are inferred from the C 1s trXPD patterns (Fig. 3b). However, the distinct spectral fingerprints of neutral and charged molecules in the C 1s trXPS spectra, together with a trXPD R-factor analysis, consideration of steric constraints, and pair-potential calculations, allow us to determine the direction of rotation and disentangle the atomic rearrangement for each species."

Line 215ff: "Yet, within the sensitivity of the trXPD experiment, we observe no signatures consistent with counter-rotating domains under continuous photoexcitation."

Reviewer #3

We thank Reviewer #3 for the positive overall assessment and for the specific suggestions to improve clarity.

Fig. 3 b,c: We added red crosses to highlight the 3 and 6 peaks in the XPD. We added the gamma point in Fig. 3 a.

We have revised the figure annotations and the accompanying text to explicitly label and describe the three-fold and six-fold peak structures in Fig. 3b, making them immediately identifiable.

Even though there might be still room for improvement of the manuscript presentation (as exemplified above), the new passages of the manuscript now explain what is obtained from data and what is obtained from interpretation. I would suggest that the authors also update the abstract and the introduction in this regard. For example:

- "... we uncover the synchronized rotational motion of molecules on..." This is an interpretation of the data, right? Even though the authors report a time-resolved measurement, the synchronous motion is not directly measured.

•“ ... and the formation of homochiral domains.” Also, this is an interpretation following static LEED and tr experiment analysis.

We rephrased the introduction to:

Line 47ff: “Here, we reveal ultrafast spectroscopic fingerprints of a collective rotational response of molecules on a 2D material following photoexcitation. Our results suggest that photoinduced charge transfer reshapes the interfacial energy potential, giving rise to macroscopic, unidirectional molecular rotation and the formation of a homochiral molecular arrangement.”